# A Current Sharing State Estimation Method of Redundant Switched-Mode Power Supply Based on LSTM Neural Network

**Peng He** ⓘ**, Quan Zhou *** ⓘ**, Libing Bai, Songlin Xie and Weijing Zhang**

School of Automation Engineering, University of Electronic Science and Technology of China, Chengdu 611731, China; hpcaep@163.com (P.H.); libing.bai@uestc.edu.cn (L.B.); 201921060306@std.uestc.edu.cn (S.X.); 13708201980@163.com (W.Z.)
* Correspondence: quanzhou@uestc.edu.cn

**Abstract:** Redundant Switched-mode Power supplies (SMPSs) are commonly used to improve electronic systems' reliability, and accurate estimation of the current sharing state is significant for evaluating the system's health. Currently, the current sharing state estimation is mainly realized by using current sensors to detect each branch's current, and the deployment and maintenance costs are high. In this paper, a method for power supply current sharing state estimation based on LSTM recurrent neural network is proposed. By taking advantage of subtle differences in the inherent spectral characteristics of SMPSs, this method only needs to detect the voltage ripple at the switching frequency of the load terminal to estimate the output current of each power supply branch. The verification experiment on the three-power redundant experimental platform shows that the estimation error is less than 10%. The method has the characteristics of simple structure, non-invasion, convenient deployment and maintenance, so it has high application and promotion value.

**Keywords:** switched mode power supply; current sharing; ripple wave; spectrum signature

## 1. Introduction

Electric power, oil, natural gas, medical and other fields have extremely stringent requirements on the reliability of power supply systems [1]. The diode-based parallel redundant power supply system without current sharing control has a simple structure, high reliability, low cost, and easy expansion, so it is a mainstream method to improve the reliability of power supply systems. However, the current sharing state significantly impacts the reliability and remaining useful life of these systems. According to the research of Phoenix company, in the case of uneven current, the temperature of a heavy-load power supply will be more than 10 degrees higher than that of a light-load power supply, and the average lifespan will be about 50% lower [2]. On the other hand, the non-current sharing state of the parallel power supply will also affect the power supply parameters of the power supply input terminal, increase the harmonics of the system, and cause the system performance to decline [3]. Therefore, online monitoring of the current sharing status of redundant power supplies can timely find out the unbalanced system load caused by various factors such as installation, temperature drift, and component parameter changes [2]. It plays a significant role in maintaining the redundant power system [4].

At present, the current sharing state detection methods mainly depend on each branch's current measurement [5], which is based on the resistor or inductor connected in series in the branch [6–8]. These methods require multiple current sensors and a complex connecting structure to evaluate the detection results. These disadvantages make the current state detection methods challenging to implement. In order to realize the current sharing state estimation under single-point detection, the researchers pioneered research on ripple spectrum analysis of the load terminal voltage [9–11]. However, this method requires changing the operating mode of the SMPS to encode the current sharing information in the ripple of the output voltage and sometimes even requires that the SMPS

work in discontinuous conduction mode. This change inevitably deteriorates the power supply's output characteristics and design complexity.

Theoretical studies show that the harmonics of SMPS are related to the conversion efficiency [12,13]. The conversion efficiency is also different under different load conditions, so the harmonics of the power supply are related to the output power of the power supply [14]. Spectral fingerprints can be formed based on the unique inherent spectral characteristics caused by individual differences in the SMPSs. In parallel mode, the spectrum of each power supply is superimposed on each other, and the spectrum change of the ripple signal at the load end must contain the information of the current of each branch. Therefore, this paper designs a spectral fingerprinting algorithm based on LSTM recurrent neural network to effectively extract the branch's current information in redundant SMPS systems. Experiments on the three-power redundant experimental platform show that this method's estimated error of the branch power supply is less than 10%.

## 2. The Spectrum Signature of SMPSs

### 2.1. Spectral Characteristics of Single SMPS

A typical AC/DC SMPS topology is shown in Figure 1, which is mainly composed of two parts: power factor correction circuits (PFC) and DC/DC modules. The electronic switch is a critical component in both of them. Therefore, the impulse generated by switching on and off is one of a power supply's primary sources of output ripple [15]. In the PFC part, the duty cycle of the control PWM signal changes within the same AC signal cycle, while in the DC/DC part, the duty cycle of the control system signal is constant. Since they are all periodic square wave signals, the spectral distribution is discrete at any static power, and there are only odd harmonics [12,15].

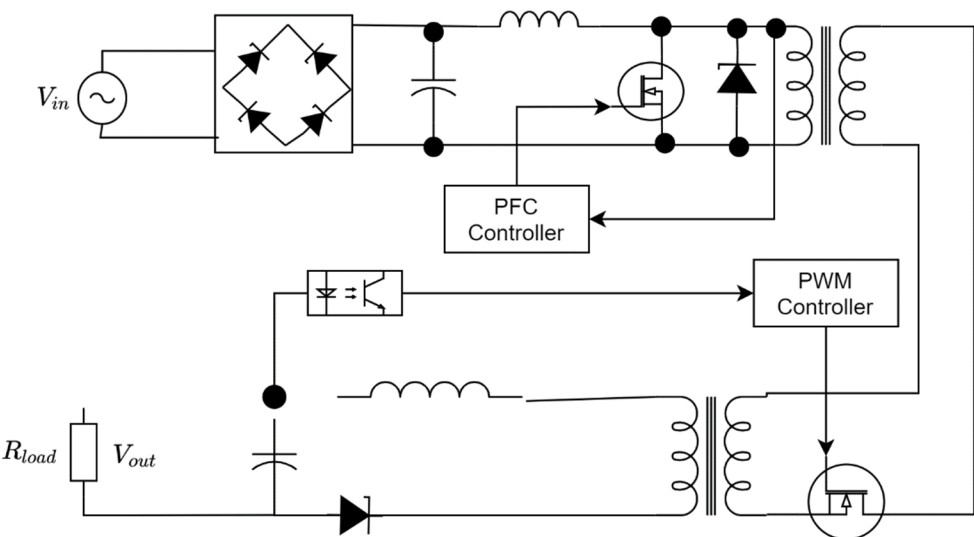

**Figure 1.** The topology of a typical SMPS.

According to the current continuity in inductor coils, the operating mode of a SMPS can be divided into two basic modes: continuous current mode (CCM) and discontinuous current mode (DCM) [12]. The advantage of CCM mode is that it is suitable for high current loads, and the output ripple is small, while the disadvantage is that the conversion efficiency is low. The DCM mode has relatively higher conversion efficiency, but the output ripple is large and is suitable for application scenarios with small current loads. In order to adapt to these two load states, the mainstream SMPS will automatically switch between two modes according to the load conditions to achieve the best balance state at each load power point. The switching frequency can theoretically be fixed in either mode, and the output power can be adjusted only by adjusting the PWM pulse width. However, this method will cause the power supply to send a large amount of electromagnetic radiation to

the outside near the switching frequency point, causing serious EMI problems. Therefore, most SMPSs usually adopt the spread spectrum method. By changing the PWM signal's frequency, the switching signal's interference ripple is dispersed into a boarder spectrum area, reducing the average electromagnetic radiation energy at the frequency point of the switching signal [16]. In unit time, the actual on-time of the PWM signal with frequency f can be expressed as:

$$t_{on} = \frac{1}{T_f}\left(T_f * R_{duty} - T_{rise} - T_{fall}\right) = \left(R_{duty} - \left(T_{rise} + T_{fall}\right)f\right) \tag{1}$$

where $T_f$ is the period of the signal, $T_{rise}$ and $T_{fall}$ are the turn-on rise and fall times, respectively, and $R_{duty}$ is the duty cycle. It can be seen from the above formula that when the duty cycle is constant, with the increase of the frequency $f$, the actual on-time decreases, and the output power is also smaller. On the contrary, the smaller the frequency, the larger the output power. At the same time, each time the switch is switched, there will be a short-term pulse impact, so the higher the pulse frequency, the more times the switch is switched, and the greater the accumulated interference energy. Therefore, in the output ripple, the spectral change around the switching frequency is shown in Figure 2.

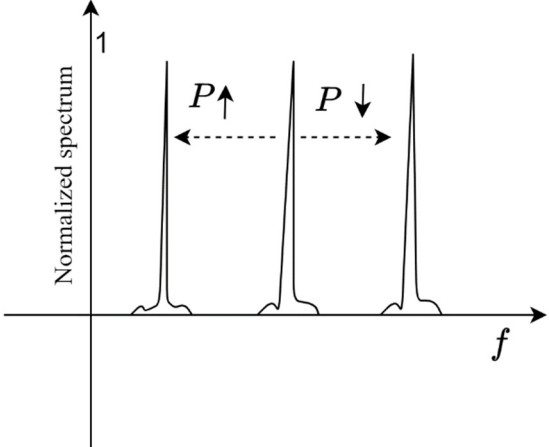

**Figure 2.** The relationship between switching frequency and output power.

As can be seen from the figure, the switching signal frequency and its amplitude in the output ripple signal are negatively correlated with power. Since the transmission line seriously affects the ripple amplitude in the redundant power supply circuit system, the frequency can remain unchanged. The following relationship can express the relationship between the frequency and the output power.

$$P(t) = F(f(t)) \tag{2}$$

where $f(t)$ represents the frequency change of the ripple signal, $P(t)$ represents the output power change corresponding to the power supply, and F represents the functional relationship between the output power and the ripple amplitude and frequency.

On the other hand, because the power switch, input filter capacitor, inductor and other devices have subtle parameter differences in each power supply, the pulse interference signal generated at the moment when the switch is turned on and off will also be different. These differences makes the shape of the ripple spectrum of different SMPS units slightly different under the same working conditions [17]. The frequency spectrum near 200 Khz switching signal frequency point of three power supplies of the same type (Phoenix QUINT-PS-100-240AC/DC/10) is shown in Figure 3, under the same input conditions (220 V, 50 Hz), the same output (24.1 V), the same load (15 Ω cement resistance). It can be seen that the overall shape of the spectrum of three power sources is the same, but in the red box

with the amplitude close to 0, the sector of the No. 2 power source is obviously smaller than that of No. 1 and No. 3, while No. 3 has the largest sector. This subtle difference allows the ripple spectrum to be used for branch power identification in redundant systems.

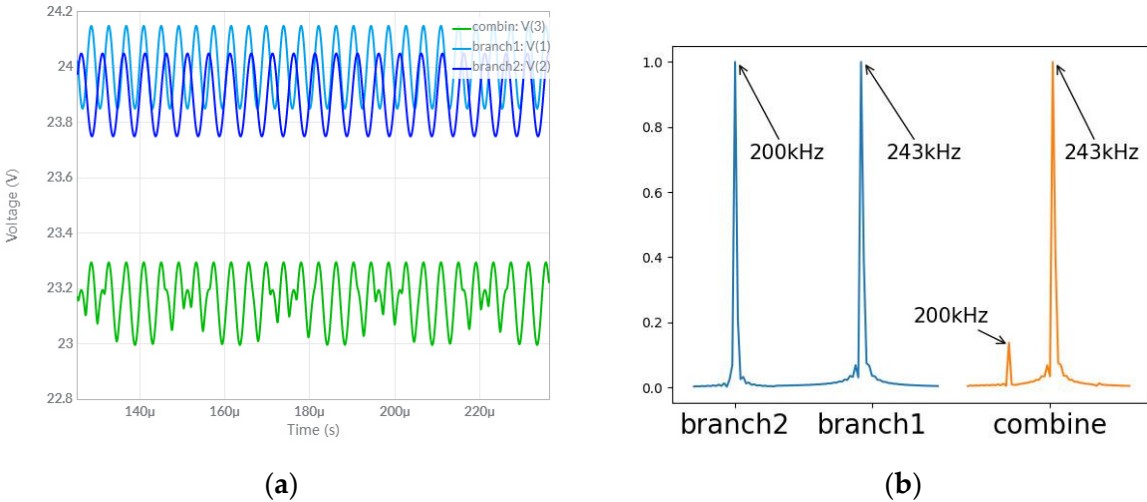

(**a**)  (**b**)

**Figure 3.** (**a**) Voltage waves in a diode-based simple parallel redundant system with two SMPSs; (**b**) The spectrum of three voltage waves.

However, in practical application scenarios, the output power of SMPSs changes frequently and dynamically, and the switching of the spectrum also changes with the change of the output power. These changes make it difficult for traditional waveform recognition and image recognition methods to complete the above tasks. Therefore, this paper introduces LSTM neural networks and uses the spectrum as a time-series signal to identify the waveform of the spectrum to solve this problem.

### 2.2. Spectral Superposition Characteristics of Diode-Based Parallel Redundant SMPS System

The diode-based simple parallel redundant SMPS system utilizes the unidirectional conduction characteristic of diodes, which also make the output power of the branch power supply closely related to the output voltage of all branch power supplies. Since the power supply is affected by factors such as the use environment, working conditions, and component ageing, the output voltage of SMPSs will change slowly over time. The variation of the output voltage of the branch power supply leads to the current uneven state of redundant power supply systems. For an ideal SMPS, the output voltage is constant, and there is no ripple, even if the voltage difference of mv level will cause a severe uneven current state of the system. For actual SMPSs, this situation is improved due to ripple. For any SMPS, its output voltage signal can be expressed as:

$$v(t) = A_0 + \sum_{i=1}^{N} A_i \sin(w_i t + \varphi_i(t)) \tag{3}$$

where $v(t)$ is the output voltage, $A_0$ is the DC component of the output voltage of the SMPS, and $A_i \sin(w_i t + \varphi_i(t))$ is the $i$-th ripple component. Due to the existence of the ripple component, the output voltage of the power supply actually changes within the range of $[A_0 + A_{min}(t), A_0 + A_{max}(t)]$, where $A_{min}(t)$ and $A_{max}(t)$ are the synthesized ripple components Minimum and maximum values. When the variation range of the output voltage of each branch power supply has overlapping areas, the output voltage of the redundant system is related to these branches, and the output power of the outgoing power supply is also related to the degree of overlap between the sections. In order to simplify the analysis process, the dual power supply redundancy system is taken as an example. Assuming that there is only one sinusoidal ripple component whose initial phase

is 0 in the output voltage of each branch power supply, the output voltages of branch 1 and branch 2 can be expressed as:

$$v_1(t) = A_{01} + A_{11}\sin(w_{11}t + \varphi_{12}(t)) \tag{4}$$

$$v_2(t) = A_{02} + A_{12}\sin(w_{12}t + \varphi_{12}(t)) \tag{5}$$

Then the voltage variation range of branch 1 is $|v_1(t)| = [A_{01} - A_{11}, A_{01} + A_{11}]$, and the voltage variation range of branch 2 is $|v_1(t)| = [A_{02} - A_{12}, A_{02} + A_{12}]$. If $|v_1(t)| \cap |v_2(t)| \neq \varnothing$, the output voltage of the redundant power supply is determined by the maximum value of the instantaneous voltage of branch 1 and outgoing 2 voltages at time $t$, which can be Expressed as:

$$v_o(t) = max\{v_1(t), v_2(t))\} \tag{6}$$

Since the output DC component $A_{0i}$ of each branch power supply, the initial phase of the ripple $\varphi_i(t)$ and the ripple frequency $w_{1i}$ are different, the output voltage of the redundant power supply is formed by the competition between the two outgoing paths in the overlapping interval. Figure 4 shows that when the power outputs of branches 1 and 2 are $v_1(t) = 24 + 0.15\sin(243 * 2\pi t)$ and $v_2(t) = 23.9 + 0.15\sin(200 * 2\pi t)$, respectively, the double Simulation results of the output voltage of a parallel redundant system of power supplies.

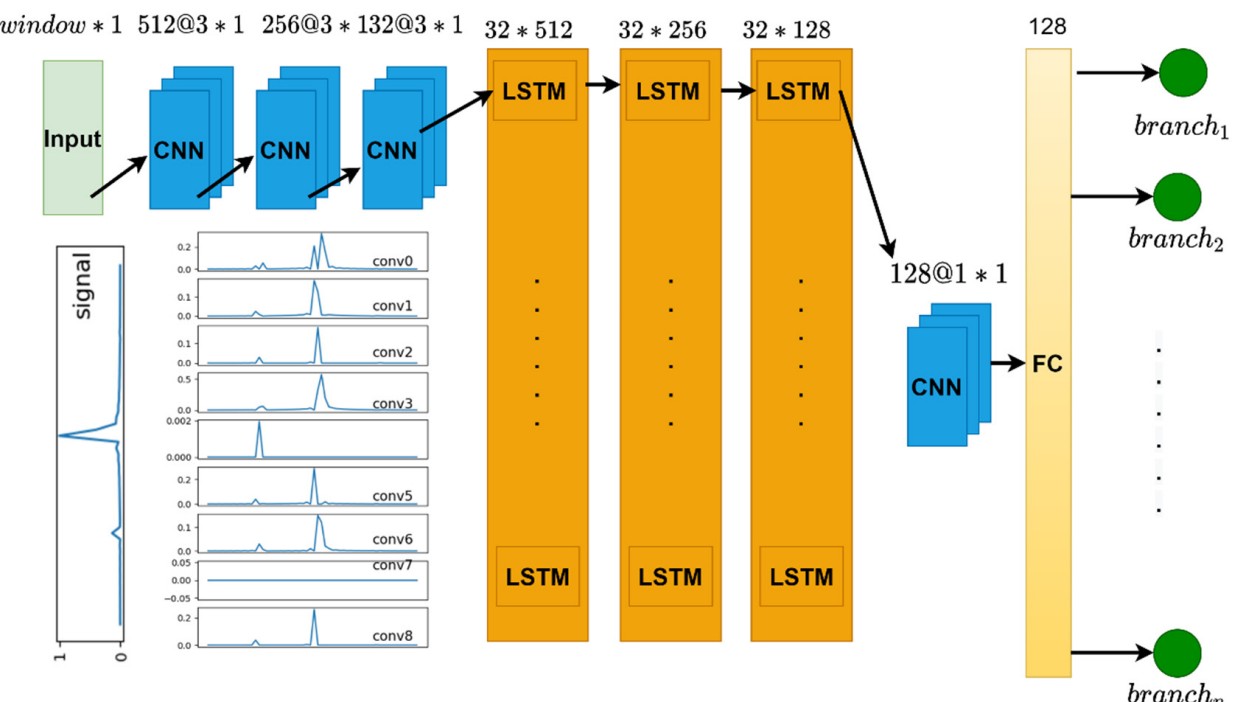

**Figure 4.** The proposed LSTM model.

As can be seen from the figure, the output voltage of a redundant system consists of voltages of branch 1 and branch 2. Since the voltage of branch 2 is relatively small, it accounts for a relatively small proportion of the overall output signal. The above signal is subjected to spectrum analysis, and the result is shown in Figure 5. As can be seen from the figure, the frequency spectrum of the output voltage consists of two frequency points, branch 1 and branch 2, and the amplitude of the ripple frequency corresponding to branch 2 is quite different from that of branch 1, which is different from that in the time domain. The signal ratio is the same.

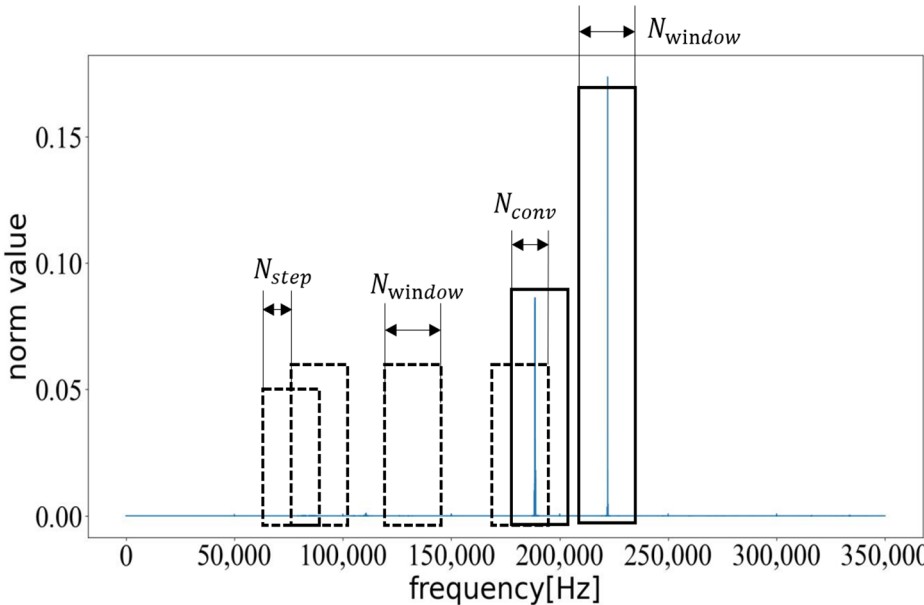

**Figure 5.** The defines of the sliding windows.

For the actual output voltage of a power supply, the ripple contains many harmonic frequency components. The amplitudes and initial companions of these harmonics are different, making the instantaneous amplitude of a power supply's output voltage have strong randomness in the time domain. Mathematical modelling of this situation can be simplified to a mathematical model in which two sets of random variables take extreme values. Assume that the output voltage of each branch power supply obeys a normal distribution:

$$v_i \sim N(A_{0i}, A_{1i}) \tag{7}$$

Among them, $A_{0i}$ is the output DC component of the *i-th* branch, and $A_{1i}$ is the variance of the voltage change. Assuming $A_{01} \geq A_{0i}, i \geq 2$, the redundant power output can be expressed as:

$$v_o(t) = a_1(t)v_1(t) + \sum_{i=2}^{n}(a_i(t)v_i(t) + (1 - a_i(t)v_1(t))) \tag{8}$$

where $a_1(t)$ is 1 when the output voltage of $v_1$ is not in the overlapping area, otherwise it is 0, specifically:

$$a_1(t) = \begin{cases} 1, & v_1(t) \notin U_1(t) \\ 0, & v_1(t) \in U_1(t) \end{cases} \tag{9}$$

where $U_1(t)$ is the part that overlaps with other branch supply voltages. For the overlapping region, the output voltage is determined by the branch voltage competition result, so

$$a_i(t) = \begin{cases} 1, & v_i(t) \geq v_j(t) \\ 0, & v_i(t) < v_j(t) \end{cases} \quad j \geq 1, j \neq i \tag{10}$$

In the overlapping area, the voltages of each branch compete fairly, and the result of the competition is related to the proportion of the voltage of each branch in the overlapping area of branch 1. Taking the dual power supply redundant system as an example, the proportion of each branch in the redundant output voltage in the overlapping area is 1/2, so the proportion of the voltage component of each branch in the redundant output voltage is:

$$\frac{P(v_0(t) = v_1(t0))}{P(v_0(t) = v_2(t0))} = \frac{P(v_1(t) \notin U_1(t)) + \frac{1}{2}P(v_1(t) \in U_1(t))}{\frac{1}{2}P(v_1(t) \in U_1(t))} \tag{11}$$

It can be seen from the above formula that in the diode-based parallel redundant system, the total output voltage includes the voltage output components of each branch, and the proportion is related to the overlapping area of the actual output voltage range of the branch. At the same time, in this process, the redundant output voltage signal can be regarded as the signal obtained by recombining the power supply voltage of each branch after sampling. Based on the above characteristics, the spectral information of each branch power supply can be extracted from the total output ripple of a redundant power supply.

## 3. Proposed LSTM Model

According to (8) and (11), it can be seen that the total output ripple of a redundant system is composed of the power supply ripple signal of each branch in a particular proportion, so in the total output ripple spectrum, the current sharing status of the system can be reflected by the amplitude of the spectrum of each branch's ripple. However, there are the following problems in practical applications. First, the amplitude of the SMPS ripple itself is small at the mv level. Its size is related to various factors such as output power, line impedance attenuation, space interference, etc., which makes the ripple. The amplitude of the voltage cannot accurately reflect the output power of the branch circuit; secondly, the relative proportional relationship of the amplitude of the ripple voltage cannot accurately judge the current sharing state of the system. For example, in a dual power supply system, when the voltage of one of the branch power supplies is too low, the redundant output does not include the outgoing voltage. Since the amplitude-based method itself cannot accurately calculate the actual power of the branch power supply, it cannot distinguish the current sharing state of the system from the single power supply state.

According to the ripple spectrum characteristics of the SMPS obtained in I, this paper mainly realizes the identification of the current sharing state of the diode-based parallel redundant power supply based on the frequency characteristics of the ripple. Since the ripple spectrum changes dynamically with the load, and the shape difference of the spectrum is relatively subtle, it is difficult for traditional image processing and feature recognition methods to solve this problem. Spectral and voice data have the same dimension and shape and are typical time-series data. The spectral data of the ripple in the diode-based parallel redundant SMPS system, compared with the general timing and spectral data, is mainly characterized in that there are only slight differences in the spectral shape of each power supply and the frequency occupied by these differences The point range is minimal relative to the entire spectral space.

In order to identify each branch power supply from the slight difference in the frequency spectrum, the corresponding neural network model is designed in this paper. As shown in the Figure 6, the model mainly strengthens the microwave difference identification from two aspects.

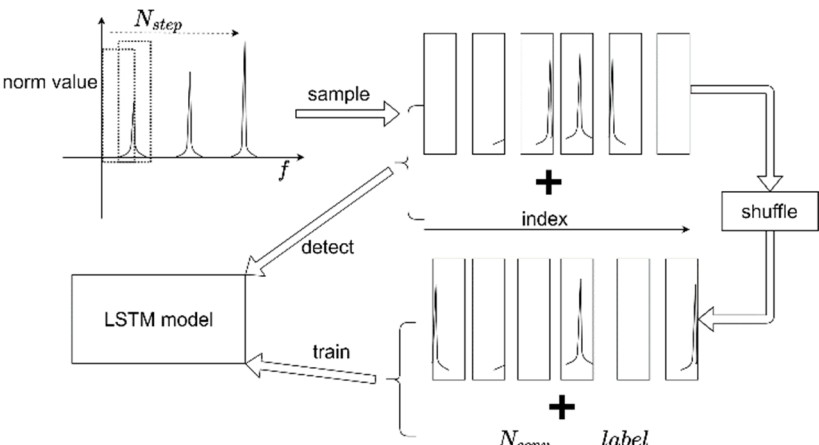

**Figure 6.** The training and detection topology.

1: Currently, RNN-like neural networks are one of the most efficient methods for processing such data [18,19]. As a special RNN, LSTM is characterized by the addition of three gates, which mainly solve the problems of gradient disappearance and gradient explosion in the process of long sequence training. Although the length of a single spectral data itself is limited, in the ripple spectrum, there are only slight differences in the spectral shape of each power supply, and the frequency range occupied by these differences is very small relative to the entire spectral space. For these small differences, the overall spectral sequence is a longer time series. Therefore, the use of LSTM can avoid these small differences being replaced by interference factors during the training process.

2: CNN plays a very good role in extracting subtle features of signals [18], so a CNN network layer is added to the front stage of LSTM, and different convolution kernels are used to amplify and extract the tiny signal features in the spectrum, and then send them to the LSTM network for training. identify.

In addition to identifying the branch power supply based on the ripple spectrum, it is also necessary to identify the output power of the branch power supply to detect the current sharing state of a redundant system. According to the linear relationship between the output power of a power supply and the switching frequency, the actual output function of the power supply can be inverted through the center frequency of the waveform in the spectrum. The LSTM network model can only classify the ripple spectrum, and it is difficult to calculate the center frequency corresponding to the spectrum. In order to solve this problem, this paper proposes a data acquisition and processing method based on a sliding window. The center frequency is estimated by the position of the window, so that the designed neural network model can not only identify the branch power supply, but also realize the branch circuit. An estimate of the actual output power of the power supply.

The design of the sliding window is shown in the figure, where the window length $N_{window}$ is determined by the maximum width of the switching frequency ripple spectrum of all SMPSs in the system; the window sliding step $N_{step}$ satisfies the condition $1 \leq N_{step} \leq N_{window}$, when the $N_{step}$ is smaller, the sliding The more times, the higher the resolution of frequency positioning, the more accurate the estimation of the actual output power of the power supply, but at the same time, the more the number of windows, the greater the amount of calculation required; otherwise, the amount of calculation would be reduced, and the estimation accuracy would be lower. When $N_{step} < N_{window}$, there will be incomplete overlap between the sliding window and the ripple spectrum waveform window. In order to avoid introducing signals with non-ripple spectrum features into the training process and affecting the training results, the window coverage parameter $N_{conv}$ is defined. The value of this parameter is the number of coincident data points between the sliding window and the label window. Only when $N_{conv} > a * N_{window}$, the sliding window is marked with the same label for training, otherwise the data label of the window is defined as empty. Among them, the threshold coefficient is $0 < a \leq 1$. When $a$ is larger, the number of labeled windows is smaller, and the training accuracy is higher, and the convergence is easier, but the amount of original data required is larger; otherwise, the smaller the amount of original data required. In addition, the accuracy will decrease, so this parameter needs to be balanced according to the actual training scenario.

Since the position of the sliding window needs to be marked to invert the actual power of the power supply, the entire training and detection process needs to be adapted accordingly. The adaptation method is shown in the figure:

The overall identification method process mainly includes three parts:

1. Data preparation:
    (a)  Collect the ripple signal at the load end and perform spectrum transformation, and extract the spectrum data S in the window of the switching spectrum variation range;
    (b)  Adjust the output voltage of the SMPS of the *i-th* branch so that its actual output power decreases, while the actual output power of other branches increases. Therefore, there will be a frequency component waveform shifted

to the left in the frequency spectrum, while the other frequency component waveforms are shifted to the right. From this, it can be determined that the left-shifted frequency component is the spectral waveform corresponding to the *i-th* branch;

(c)　Repeat step b until the spectral components corresponding to each branch power supply are determined;

(d)　Adjust the load resistance to make each power supply work under different output power conditions, collect ripple spectrum data at the same time, and label each frequency component waveform in the spectrum data. For the coincident spectral components, count the number m of spectrum quantities, and the standard is the coincident power supply m;

(e)　Repeat step d to realize the collection of data sets under different working conditions;

(f)　Adjust the power supply voltage of a certain branch to change the overall current sharing state of the system, and then repeat steps d to e to sample and label the spectrum data in the new current sharing state;

(g)　Repeat step f to complete data collection for different current sharing states.

2.　model training

(a)　Count the maximum width of the spectrum in the data set, determine the sliding window parameter $N_{window}$, step $N_{step}$, coverage threshold coefficient a, etc.

(b)　The spectrum data is divided and extracted according to the sliding window parameters, and the sliding coordinate index corresponding to each interception window, the coverage $N_{conv}$ and other information are recorded in the extraction process;

(c)　Randomly scramble the window data set and send it to the LSTM neural network model for training;

3.　actual detection

(a)　Collect the ripple signal at the load end, perform spectrum transformation, and extract the spectrum data S' in the window of the switching spectrum variation range;

(b)　Use a window of length $N_{window}$ and step $N_{step}$ to intercept spectral data. The intercepted window data is sent to the trained LSTM neural network model for identification. For non-empty identification results, use the sliding coordinates of the window to calculate the actual output power of the branch power supply;

(c)　According to the number of identified branches and the output power of the branch power supply, the current sharing state of the overall system is judged.

## 4. Experimental Results

To verify the effectiveness of this method, a diode-based three-electrical parallel redundant system test platform is built. The framework of a test system platform is shown in Figure 7.

The experimental power supply model is phoenix QUINT-PS-100-240AC/DC/10, the rated input of this power supply is AC 100 V to 240 V, the rated output is 24 V, 10 A, and its secondary side switching frequency varies with the load between 180 kHz and 260 kHz, and It has a voltage adjustment screw, which can adjust the power outlet voltage in the range of 23.5 V to 24.5 V. Three identical power supplies are connected in simple parallel redundantly via diodes (40 A, phoenix QUINT-DIODE/40). At the same time, in order to verify the experimental results, the voltage and current on each branch are collected synchronously. The data acquisition card is NET-2991, which has 16 synchronous analog acquisition channels, and the acquisition speed of each channel is 1 MS/s. The data is sent to the PC through the thousand M network port for storage and analysis. The load end uses a 100 W cement resistor, the resistance value is from 6 Ω to 10 Ω, and the simulation of load changes is realized by series and parallel. According to the data acquisition and

training steps described in V, the power supply ripple at both ends of the cement resistor was collected. Among them, the resistance simulates the resistance value change from 3 Ω to 8 Ω with an interval of 1 Ω, and the voltage modules of each branch of three power supplies vary from 23.5 V to 24.5 V with an interval of 0.2 V. Under the same working conditions, data collection and labeling are carried out for at least 3 min and every 30 s, and more than 3600 sets of spectrum data are collected in total. For the first group of spectrum data, the frequency of the switching frequency band is intercepted as the original data, that is, the spectrum data in the range of 180 kHz to 260 kHz. According to statistics, the widest switching frequency ripple spectrum width is 20 Khz, so the window size is 20,000. In order to estimate the actual output power of the outgoing power supply as accurately as possible, the sliding step size is 1/20 of the window size, that is, 1000. Since the step size is small, a larger threshold value a = 0.8 is selected to reduce the influence of noise signals. Finally, these labeled data are divided into training set and test set according to 3:1, and are put into the model for training. The training result is shown in Figure 8.

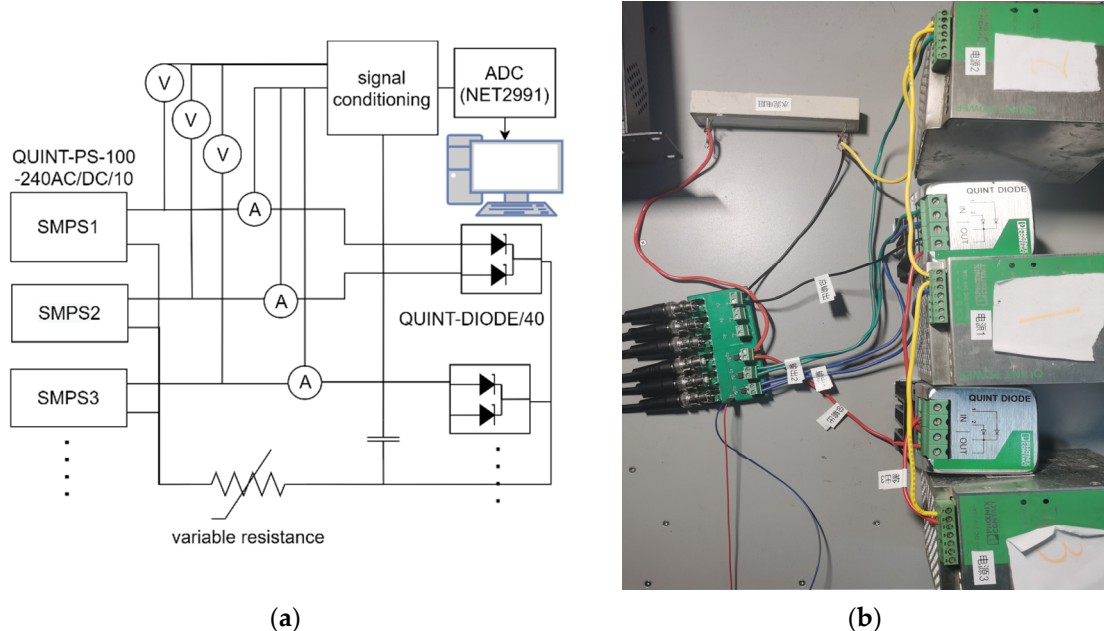

| (**a**) | (**b**) |

**Figure 7.** (**a**) the structure of the experiment platform; (**b**) the experiment platform.

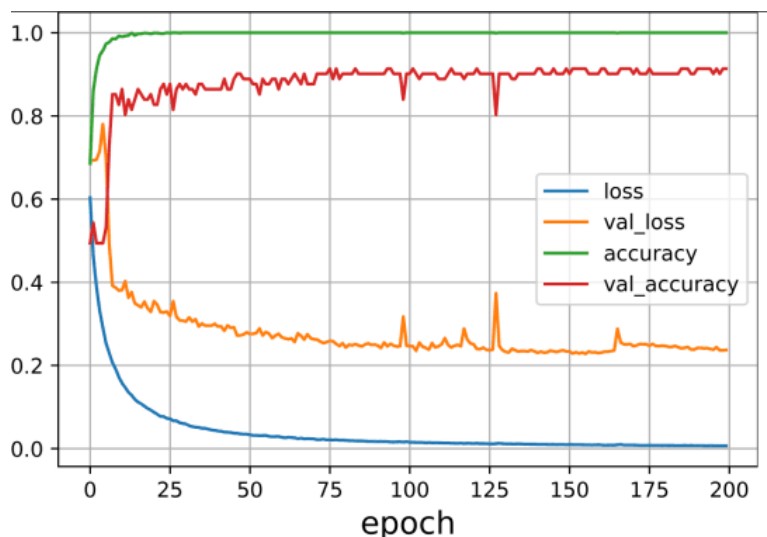

**Figure 8.** The training process.

The training results can reach a state of convergence within 100 iterations, and the accuracy rate can reach more than 96%. The training model can also achieve a recognition accuracy of more than 92% on the test dataset. Finally, the model is deployed, and the real-time redundant current sharing state identification effect of the three-power redundant platform is performed according to the actual detection steps described in V as shown in the figure.

An application demo is built with the posted method for monitoring the current state of the redundant power system used in the experiment. The real-time ripple wave, its spectrum and the output power of each branch are shown in the application as Figure 9. In the figure, two RMBSs are identified with two windows in the middle part, and the actual output powers of each power are estimated in the top left. The result we can see is that power 1 is outputting 23% more power than power 2, and the current sharing state is not good for the health of the power system.

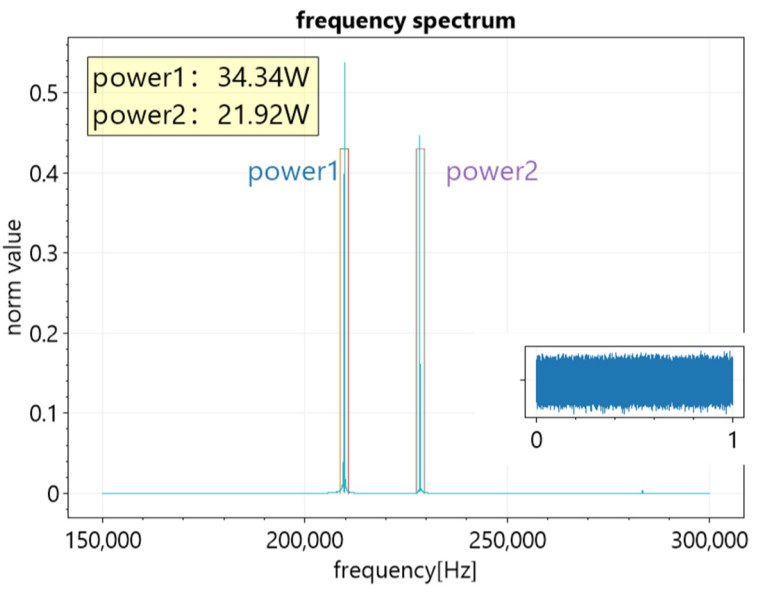

**Figure 9.** The current Sharing state estimation result.

## 5. Conclusions

In this paper, a novel current sharing detection method based on LSTM is proposed for parallel-connected SMPS system. The method is based on the neural network structure of CNN + LSTM to identify the subtle differences of each branch's RMBS in the ripple spectrum.In addition, a sliding window algorithm is added to estimate the actual output power of each SMPS. The method's feasibility is verified through the experiments and the estimation accuracy is acceptable for application. Compared with traditional methods, the most prominent advantage is that the posted method only needs to detect the ripple voltage at the load end to detect the current sharing state of the parallel redundant system, and the actual output power and current of each branch can be estimated. Therefore, this method has the advantages of low cost, non-invasiveness, and easy implementation.

At the same time, there are two main problems in practical applications worthy of further research.

1: The posted method cannot currently distinguish CCM and DCM mode of SMPSs. That is to say, its application needs to be established on the premise that all SMPSs work in a single certain mode. This limitation reduces the application value of the posted method.

2: Our experiments are based on resistors, and the interference factors of the same frequency band introduced by the load terminal are not considered. It will be a huge challenge in practical applications.

**Author Contributions:** Conceptualization, P.H. and Q.Z.; methodology, L.B.; software, S.X.; valida­tion, W.Z.; formal analysis, P.H.; investigation, P.H.; resources, S.X.; data curation, S.X.; writing—original draft preparation, P.H.; writing—review and editing, Q.Z.; visualization, S.X.; supervision, L.B.; project administration, L.B.; funding acquisition, L.B. All authors have read and agreed to the published version of the manuscript.

**Funding:** This work was supported by the National Natural Science Foundation of China under Grants No. U1830133 (NSFC) and the Project of Sichuan Youth Science and Technology Innovation Team, China (Grant No. 2020JDTD0008).

**Institutional Review Board Statement:** Not applicable.

**Informed Consent Statement:** Not applicable.

**Data Availability Statement:** Not applicable.

**Conflicts of Interest:** The authors declare no conflict of interest.

## Nomenclature

| | |
|---|---|
| SMPS | Switched-mode Power Supply |
| LSTM | Long short-term memory |
| AC/DC | Alternating Current/Direct Current |
| PFC | power factor correction circuits |
| $t_{on}$ | the actual on-time of the PWM signal |
| $T_f$ | the period time of the signal with frequency f |
| $R_{duty}$ | Duty cycle |
| $T_{rise}$ | turn-on rise time |
| $T_{fall}$ | turn-on fall time |
| $f$ | Signal frequency |
| $P(t)$ | Output power at time t |
| $F(f(t))$ | The relationship function between frequency and output power |
| $v_i(t)$ | Output voltage of the SMPS in the *i-th* branch at time t |
| $U_{i(t)}$ | the overlap part between the output voltage of the *i-th* branch and other branches' |
| $a_i(t)$ | The factor weight of the *i-th* branch's voltage at the load side |
| $A_{ij}$ | The maximum amplitude of the *i-th* harmonic of the *j-th* branch SMPS |
| $N$ | Integer which is theoretically infinite |
| $\omega_{ij}$ | Angular frequency of the *i-th* harmonic of the *j-th* branch SMPS |
| $\varphi_{ij}$ | Phase shift of the *i-th* harmonic of the *j-th* branch SMPS |
| $P$ | Power of the signal |

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
