# Peer review of "A Current Sharing State Estimation Method of Redundant Switched-Mode Power Supply Based on LSTM Neural Network"

_applsci, doi:10.3390/app12073303_

Round 1

Reviewer 1 Report

The paper deals with the problem of estimating the current sharing state in Redundant Switching Power Supplies. Previous related efforts have relied on simple techniques such as using resistors or inductors connected in series, requiring complex structures, or through ripple spectrum analysis that require changing the operating mode of the switching power supply. On the other hand, this paper proposes a novel spectral finger-
printing algorithm based on a combination of convolutional neural networks and Long short-term memory recurrent neural networks. This method provides the added advantage of being self adaptive to the characteristics of the power supplies, making it highly dynamic due to its ability to learn. However, the same advantage its it major shortcoming and open question: how well does the method adapt to changing features of the load, but not only in terms of resistance, but also capacitance and inductance. In other words, once the neural network has learn a model, it can be used in the power supply. Buy the model estimation accuracy depends on the training data, which in this case is based on a 100W load of 6Ω to 10Ω. How will the technique behave under different load conditions? Is it necessary to re-train the neural network with different loads?

In addition the paper has some minor writing issues:

- Revise how "the" is used through the paper
- "state detection methods are challenging to implement." -> "state detection methods challenging to implement."
- "the energy of each branch ripple spectrum can be reflected current sharing status of the system.": revise/rewrite
- "and the output power is accurate lower sex.": revise

Author Response

Thanks for your comments which are very instructive for our work.

1:how well does the method adapt to changing features of the load, but not only in terms of resistance, but also capacitance and inductance. In other words, once the neural network has learned a model, it can be used in other power supplies.

Reply:  The capacitance and inductance only theoretically affect the switching frequency's absolute amplitude in the frequency spectrum, but not the frequency. On the premise that there is no other harmonic interference with the same frequency band (180kHz~260kHz) introduced, the ratio between the switching frequency and different frequency components will keep steady. The posted method is based on the normalized spectrum,  which can eliminate the influence of absolute amplitude. Therefore, the inductive and capacitive changes at the load end will not affect the output performance of the method. Of course, the application environment of the industrial field is extremely complex. In the case that the interference of the same frequency band cannot be excluded, more research needs to be done for this problem. Recently, we are building a new platform that contains PLCs, sensors, signal routers etc., for further research.

   we add a brief description of the inadequacy for this problem in conclusion at line 418~426

2: The model estimation accuracy depends on the training data, which in this case is based on a 100W load of 6Ω to 10Ω. How will the technique behave under different load conditions? Is it necessary to re-train the neural network with different loads?

Reply: In the experiment, multiple 100W resistors are used in series and parallel to achieve a load change from 3Ω to 8Ω. Therefore, this method is effective from 30% to 80% of the rated output power of a single power supply. Currently, our method cannot distinguish the two working modes based on the frequency spectrum alone, the experiment is only for the case of more than 30% of the rated power because the working mode conversion of CCM and DCM occurs near 30% of the rated power.

Since the load that is too high or too low will affect the output performance and life of the power supply, the power system design usually ensure that the power supply can work at 40% to 80% of the rated power for a long time. Our experiments cover this condition and can meet the needs of applications。

In conclusion, this method needs to be retrained when the load change is less than 30%. For the case of including two working modes, this method cannot be adapted at present, which is also a direction of our further research. A potentially viable solution for this problem is tracking the frequency change trajectory. The design and verification of related experiments are currently underway.

we add a brief description of the inadequacies for this problem in conclusion at line 418~426.

3: Revise how "the" is used through the paper

Reply: Some ‘the’ in the paper are changed through the paper with the help of professionals.

4: "state detection methods are challenging to implement." -> "state detection methods challenging to implement."

Reply: It has been made in the manuscript in line 46.

5: "the energy of each branch ripple spectrum can be reflected current sharing status of the system.": revise/rewrite

Reply: It has been rewritten as “the current sharing status of the system the energy of each branch ripple spectrum can be reflected by the amplitude of the spectrum of each branch’s ripple.” in line 219~221.

6: "and the output power is accurate lower sex.": revise

Reply: The sentence has be changed as: “the amount of calculation would be reduced, and the estimation accuracy would be lower” in line 282-283.

Reviewer 2 Report

Congratulations on your interesting research topic. The work is of a high degree of difficulty.

The article is prepared correctly, but there were some small errors:

  1. The language used at work and its organization is very good.
  2. There are many formulas and equations in the article, not all of them have a symbol defined.Please review carefully and add the symbols.
  3. Last but not least, I think that a summary table explaining all the symbols and acronyms used in this work would be beneficial for improving its readability and comprehensibility; therefore, I suggest including this list of symbols/acronyms in a dedicated section of this paper.
  1. The introduction in the article is excellent, but in my opinion there is no presentation of the methodology for solving the problem undertaken by the team.
  2. The summary in the article is vague, there is no clear description of effects and results, which was the scientific goal of this work, please explain it.
  3. Please explain this sentence if it is about the output power of the electricity“….window coordinates, so as to estimate the actual output power of the out-391 going power supply”.
  4. Figure 9 requires a more detailed description and explanation.

Author Response

Thanks for your comments which are very instructive for our work.

1: The language used at work and its organization is very good.

Reply: Thanks!

2:There are many formulas and equations in the article, not all of them have a symbol defined. Please review carefully and add the symbols.

Reply: “Nomenclature” is added before “References” and all the symbols and their defines are listed in line 437.

3: Last but not least, I think that a summary table explaining all the symbols and acronyms used in this work would be beneficial for improving its readability and comprehensibility; therefore, I suggest including this list of symbols/acronyms in a dedicated section of this paper.

Reply: “Nomenclature” is added before “References” in line 437.

4: The introduction in the article is excellent, but in my opinion there is no presentation of the methodology for solving the problem undertaken by the team.

Reply: T The method proposed in this paper does have certain limitations. For example, it is currently impossible to distinguish the working modes (CCM and DCM) based on the spectral information alone. Therefore, the problem is solved under the premise that all the power supplies work in a certain mode. More further research is needed to solve the problem completely. At present, we are developing methods to solve this problem by tracking the change of frequency trajectory and using preprocessing to amplify the personalized differential features of the ripple of each power supply.

For the limitations of the current method, a corresponding note has been added to the conclusion at line 418-427.

5:The summary in the article is vague, there is no clear description of effects and results, which was the scientific goal of this work, please explain it.

Reply: we rewrite the whole conclusion in line 404~427

6:Please explain this sentence if it is about the output power of the electricity“….window coordinates, so as to estimate the actual output power of the out-391 going power supply”.

Reply: The information we want to express in this sentence is that the actual output power can be estimated by the index number of the recognized window. The sentence is changed into “” in line 405.

7:Figure 9 requires a more detailed description and explanation.

Reply: A paragraph descripting Fig.9 is added after the figure in line 394~400.